# Impact of the COVID-19 Pandemic on Therapy and Outcome of Acute Exacerbations of Chronic Obstructive Lung Disease at the Emergency Department

**DOI:** 10.3390/healthcare12060637

**Published:** 2024-03-12

**Authors:** Verena Fuhrmann, Bettina Wandl, Anton N. Laggner, Dominik Roth

**Affiliations:** 1Department of Emergency Medicine, Medical University of Vienna, Währinger Gürtel 18-20, 1090 Wien, Austria; verena.fuhrmann@meduniwien.ac.at (V.F.); bettina.wandl@meduniwien.ac.at (B.W.); anton.laggner@meduniwien.ac.at (A.N.L.); 2Institute of Nursing Science, Department of Nursing Science and Gerontology, UMIT TIROL—Private University for Health Sciences and Health Technology, Eduard-Wallnöfer-Zentrum 1, 6060 Hall in Tyrol, Austria

**Keywords:** COPD, COVID-19, therapy, acute exacerbations, ED

## Abstract

This study compared the treatment outcomes of acute exacerbation of COPD (AECOPD) at an academic tertiary care emergency department before and during the COVID-19 pandemic. Analyzing data from 976 patients, our study showed a significant surge in overall respiratory therapy interventions amidst the noticeable decline in the total number of AECOPD cases during the pandemic. The marked increase in the utilization of non-invasive ventilation (NIV) was particularly important, soaring from 12% to 18% during the pandemic. Interestingly, this heightened reliance on NIV stood in contrast to the stability observed in other therapeutic modalities, including oxygen insufflation alone, high-flow nasal cannulas, and invasive ventilation. This distinctive treatment pattern underscores the adaptability of healthcare providers in the face of novel challenges, with a discernible emphasis on the strategic utilization of NIV. The shift in patient acuity during the pandemic became evident as the data showed a cohort of individuals presenting with AECOPD who were more severely ill. This was reflected in the increased use of NIV and, notably, a statistically significant rise in one-year mortality rates—from 32% before the pandemic to 38% during the pandemic (*p* = 0.046). These findings underscore the intricate balance healthcare providers must strike in navigating the complexities of patient care during a public health crisis. A closer examination of the longitudinal trajectory revealed a subtle decrease in re-admission rates from 65% to 60%. The increased reliance on NIV, a key finding of this investigation, reflects a strategic response to the unique demands of the pandemic, potentially influenced by both medical considerations and non-medical factors, such as the prevalent “fear of aerosols” and the imperative to navigate transmission risks within the healthcare setting. These insights contribute to understanding the evolving dynamics of AECOPD management during public health crises.

## 1. Introduction

The occurrence of acute exacerbation of chronic pulmonary disease (AECOPD) is a common cause for individuals seeking medical attention at the emergency department (ED) [1]. The urgency in providing rapid treatment arises from the high mortality associated with this condition [2,3]. Non-invasive ventilation support (NIV) has been established as a gold standard treatment for AECOPD over the last couple of years [3]. However, the advent of the COVID-19 pandemic in early 2020 ushered in practical challenges that impacted the implementation of this established treatment.

A great proportion of COVID-19 patients initially present with respiratory symptoms. Subjects suffering from chronic lung disease are at higher risk for severe disease progression and have an increased risk of mortality [4,5,6]. Despite the acknowledged efficacy of aerosol-forming treatments such as NIV or bronchodilator inhalation for AECOPD, a cautious approach was taken to discourage their use until the exclusion of a concurrent COVID-19 infection [6]. This is especially true for intensive or intermediate care units integrated into an emergency department (ED) with little to no means of isolating a potentially infectious patient. During the first wave of the COVID-19 pandemic, when antigen tests were not yet widely available and reliable, it took two to four hours until the results of PCR tests were available and the SARS-CoV2 status of a patient was known.

A combination of these factors could lead to a delay in respiratory therapy for patients with AECOPD.

These multifaceted challenges arising from the COVID-19 pandemic, including the potential delay in initiating respiratory therapy for AECOPD patients, were compounded by a global decrease in overall patient numbers at EDs. This was partly due to measures taken by the hospital. For example, for patients seeking care on their own (without consulting emergency medical services first), access was often restricted to individuals under long-term treatment at the hospital (e.g., patients being treated for cancer) or those referred by a specialist.

This was, in some aspects, similar to previous catastrophes such as Hurricane Sandy in 2012 and outbreaks of MERS 2015 or SARS 2003, when a significant decline in patient numbers was observed at EDs [7,8,9,10]. This effect was particularly noticeable in patients with chronic or non-urgent health problems. Those who seek help in a hospital despite a catastrophe often suffer from serious illness and require extensive testing and therapy [11,12].

The cumulative impact of these factors during the COVID-19 pandemic translated into a significant decrease in overall patient volumes seeking emergency care.

One of the key functions of an ED is the quick evaluation of patients and deciding whether they require intensive care, can be treated on a regular hospital ward, or if outpatient management is possible. In many cases, this decision can be made based on clinical presentation, medical history, and only a very limited set of necessary additional (laboratory or imaging) tests, saving precious resources. This quick-decision approach was severely impaired by the COVID-19 pandemic. Inpatient admission was dependent on PCR test results, which could take up to four hours. Treatment in the emergency department itself, especially non-invasive ventilation support, might also have been delayed. Available space was limited due to the need for isolation, and fear of an increased infection risk might have influenced the decision to start such potentially aerosol-forming treatments.

### Goals of this Investigation

Considering the limited understanding surrounding the implications of a pandemic involving an airborne virus on the care of patients grappling with respiratory insufficiency, our primary objective was to conduct a comprehensive analysis delineating the repercussions of the COVID-19 pandemic on patient flow and treatment for individuals experiencing acute exacerbation of chronic obstructive pulmonary disease.

Specifically, we analyzed data to ascertain whether there was a surge in patients necessitating comprehensive respiratory therapy such as oxygen insufflation, employment of HFNC, utilization of NIV, or invasive ventilation as a last resort. Moreover, our investigation extended beyond the immediate treatment realm, aiming to unravel the ripple effects of the pandemic on the operational aspects of the emergency department. One key aspect scrutinized was whether the pandemic triggered an extension in the boarding time experienced by patients at the ED, potentially indicative of an increased strain on the healthcare system. We probed further into whether there was a notable fluctuation in the rates of hospital admissions or intensive care unit (ICU) admissions, reflecting the systemic impact of the pandemic on the broader spectrum of healthcare services. In our quest for a comprehensive understanding, we also embarked on an exploration of longer-term consequences, delving into the potential influence of the pandemic on one-year mortality rates among patients presenting with AECOPD.

## 2. Materials and Methods

### 2.1. Study Design and Setting

We conducted a retrospective chart review investigating the temporal dynamics of the COVID-19 pandemic’s impact on healthcare delivery. The study period encompassed the initial wave of the pandemic, spanning from 15 March 2020 to 14 March 2021, and was compared with the corresponding time frame exactly one year prior, extending from 15 March 2019 to 14 March 2020. To ensure a thorough understanding of the long-term implications, all patients were followed up for a duration of one year.

In adherence to robust research methodologies, our study findings are presented in accordance with the REporting of studies Conducted using Observational Routinely collected health Data (RECORD) Statement, an extension that augments the STrengthening the Reporting of OBservational studies in Epidemiology (STROBE) statement [13]. The study was approved by our local institutional review board.

The study setting was a 2300-bed tertiary care university hospital, which is the largest of seven public hospitals in a city with 1.9 million inhabitants. The Department of Emergency Medicine is a pivotal component of this healthcare center and shoulders the responsibility for the treatment of all medical emergencies in adult patients. The department is divided into three parts, including an intensive care unit consisting of seven positions, a seven-bed intermediate care unit, and an outpatient clinic, each tailored to meet the diverse medical needs of the patient population.

Patient triage, a critical facet in the emergency care continuum, is expertly handled by qualified nurses who employ the internationally recognized Emergency Severity Index (ESI). This stratification system categorizes patients into five groups, ranging from 1 (life-threatening) to 5 (no medical resources needed) [14]. Collaborative care is a cornerstone of the department’s operational philosophy and sees physicians working in tandem with consultant physicians from various specialties such as ENT, surgery, orthopedics, and others, ensuring a thorough approach to patient well-being.

In the context of the Austrian healthcare landscape, characterized by a universal healthcare system, our study benefits from insights derived within a framework that ensures healthcare coverage for all, including hospitalization, outpatient care, and ambulance services. Noteworthy is the absence of regulations dictating the level of care, allowing patients the autonomy to choose their preferred level of care without the need to navigate through a general practitioner or any other intermediary healthcare provider.

### 2.2. Study Population and Inclusion and Exclusion Criteria

The study population consisted of all patients treated for acute exacerbations of COPD at the department during the study period, as determined by the treating physician. We decided to analyze full calendar years to allow for the known seasonal fluctuation in patient numbers and case mix. Patients who left without being seen were excluded from this study.

### 2.3. Measurements

All data were collected through the hospital’s electronic health record and retrospectively retrieved for this study.

Collected data included:Patient age and genderDate and time of arrival at the ED and leaving the ED (discharge or admission)Further management (discharge, admission to normal ward or ICU)Use of long-term oxygen therapy (LTOT) at homeVital signs on admission (heart rate, blood pressure, blood gas analysis)Laboratory findings (inflammatory signs, renal function parameters)Respiratory therapy according to the WHO ordinal scale for COVID-19 clinical research:0no clinical or virological evidence of infection1ambulatory, no activity limitation2ambulatory, activity limitation3hospitalized, no oxygen therapy4hospitalized, oxygen mask or nasal prongs5hospitalized, noninvasive mechanical ventilation (NIMV) or high-flow nasal cannula (HFNC)6hospitalized, intubation and invasive mechanical ventilation (IMV)7hospitalized, IMV + additional support such as pressors or extracardiac membranous oxygenation (ECMO)8death
Chest X-ray findings (emphysema, pulmonary infiltrates)Re-admission within one yearDeath within one year

The research team had full access to the health record database and all members were carefully trained in coding before beginning the chart review process. This pre-emptive training was a crucial step in enhancing the accuracy and consistency of data coding throughout the research project. To maintain a high standard of data quality and accuracy, the coded information underwent monitoring by the primary investigator, who conducted random checks at various stages of the process. Any observed discrepancies in the coding were promptly flagged, triggering a comprehensive review and clarification process by the research team. We used an electronic form for data abstraction. Prior to full-scale implementation, a small-scale and informal test pilot was conducted to ensure the seamless functionality and effectiveness of the electronic abstraction system. This pilot phase served as a proactive measure to identify and address any potential glitches or challenges in the system, thereby optimizing the data abstraction workflow for the main phase of the study. One year subsequent to the initial presentation at the emergency department, the research team engaged in a follow-up stage by accessing the patient’s national electronic health record once again. This revisit aimed to ascertain whether, and if so, when the patient had experienced any subsequent hospital re-admissions or had succumbed to their health challenges. This longitudinal approach to data collection not only provided valuable insights into the immediate outcomes following the emergency department visit but also enabled the exploration of longer-term health trajectories and potential correlations between initial interventions and sustained health outcomes.

### 2.4. Outcomes and Analysis

The primary outcome was the type of respiratory therapy (HFNC/non-invasive ventilation support/intubation) received by patients treated for AECOPD. Secondary outcomes included waiting time from arrival at the emergency department to admission, total number of patients treated, admitted to the ward, and admitted to the ICU for AECOPD, and proportion of patients receiving invasive and non-invasive ventilation at the emergency department.

We compared the “before” and “pandemic” groups using standard methods (the chi-square test for comparison of categorical variables and the t-test and Mann–Whitney U test for comparison of normally distributed and non-normally distributed continuous variables, respectively). We calculated absolute differences in outcomes using linear (for continuous outcomes) and binary (for categorical outcomes) regression, calculating robust 95% confidence intervals as described by Hamilton et al. We used Stata 17MP for all calculations.

## 3. Results

### 3.1. Patient Characteristics

A total of 976 patients were treated for AECOPD over the whole study period. Total visits decreased by 32% from 580 before the pandemic to 396 during the pandemic. See Figure 1 for visits per month.

There were no relevant differences regarding gender (295 (51%) male patients before and 218 (55%) during the pandemic) or age (median age 70 years; inter-quartile range (IQR) 63–77 before and 72 years (IQR 65–79) after; overall, four patients were younger than 40 years, 19 were between 40–49, 123 were between 50–59, 295 were between 60–69, 369 were between 70–79, 139 were between 80–89, and 27 were 90 years or older), nor regarding previous LTOT therapy (see Table 1). Interestingly, despite the aforementioned regulations being in place, there was also no change in means of arrival (66% walk-ins before, 65% after).

Patients were similar in both groups regarding vital signs, initial pH and pCO_2,_ and prevalence of pulmonary infiltrate, but signs of emphysema (8 vs. 12%, *p* = 0.02) and pulmonary venous congestion (14 vs. 24%, *p* < 0.001) were found more frequently in imaging studies of the pandemic group.

There was no difference in inflammation parameters at admission; mean CRP was 2.0 mg/dL vs. 1.9 mg/dL and mean white blood count was also similar (11.1 G/L vs. 11.9 G/L). Patients suffering from leukopenia (white blood count < 4 G/L) were overall a rare occurrence (2% vs. 1%). Despite a decrease (58% vs. 53%, *p* = 0.045) in patients with leucocytosis (white blood count > 10 G/L) during the pandemic, the use of antibiotics rose significantly from 18% to 23% (*p* = 0.01).

Other lab findings, including serum lactate, bicarbonate, electrolytes, creatinine, and blood urea nitrogen did not differ significantly between groups (see Table 1). Ten patients (2.6%) were diagnosed with COVID-19 [15].

### 3.2. Main Results

Comparing treatment before and during the pandemic, there was a significant increase in the number of patients requiring some form of respiratory support. This spectrum of support encompasses a range from oxygen insufflation to invasive ventilation, revealing a statistically significant rise from 71% to 80% (*p* = 0.001). Additionally, a significant rise is observed in the subset of patients receiving non-invasive ventilation support, with the percentage increasing from 12% before the pandemic to 18% during the pandemic (*p* = 0.01). Notably, however, there were no substantial differences concerning oxygen insufflation alone (49% vs. 51%), HFNC utilization (3% vs. 3%), or the application of invasive ventilation (7% vs. 8%). A detailed graphical representation can be found in Figure 2. Regarding patients diagnosed with COVID-19, six (60%) needed oxygen insufflation (WHO ordinal scale category 4), one patient (10%) received NIV and HFNC therapy (WHO category 5), respectively, and none needed invasive ventilation.

Within the emergency department (ED), a marked change is observed in the proportion of patients admitted to the department’s own Intensive Care Unit (ICU), surging significantly from 39% before the pandemic to 64% during the pandemic (*p* < 0.001). This shift underscores the evolving dynamics and heightened acuity of cases necessitating specialized critical care within the ED during the pandemic era.

Regarding further treatment after a stay at the ED, there was no change in boarding time (median 280 min, IQR 146–557 vs. 283 min, 181–515), the percentage of those who could be discharged (3.6 vs. 3.8%), those who had to be admitted to a regular ward (90 vs. 89%), or those who had to be treated at an ICU (7 vs. 8%). The length of stay experienced only a marginal increase, with a mere seven-minute difference, as the mean time at the ED extended from 6 h 33 min before the pandemic to 6 h 40 min during the pandemic (Figure 3). The mean boarding time of COVID-19 patients was 164 min.

An exploration into the longitudinal trajectory of patient outcomes reveals that within one year, 65% of patients before the pandemic and 60% during the pandemic experienced re-admission to the hospital. However, a significant shift is noted in one-year mortality rates, escalating from 32% before the pandemic to 38% during the pandemic (*p* = 0.046). This pronounced change is shown in Figure 4. The observed variations underscore the far-reaching implications of the pandemic on both the immediate and extended healthcare trajectories of patients treated for AECOPD.

## 4. Discussion

In our comprehensive study analyzing emergency department (ED) visits for AECOPD both before and during the initial wave of the COVID-19 pandemic, a pronounced and statistically significant reduction in ED visits emerged—a trend consistent with a plethora of analogous findings in the existing literature [16,17,18,19]. Interestingly, in our study, this was true, although we included only patients with AECOPD, whereas Gutovitz et al. in their analysis of ED utilization in America during the pandemic identified infectious disease and respiratory illnesses as exceptions from the general decrease in ED visits [17]. Our findings also deviate from earlier research [18], as we did not observe any discernible differences regarding gender in the context of ED visits for AECOPD.

While our investigation revealed no alterations in the relative frequency of simple oxygen insufflation, high-flow nasal cannulas, or invasive ventilation, a substantial and statistically significant increase was observed in the overall utilization of respiratory therapy, escalating from 71% to 80%. Moreover, there was a conspicuous increase in the proportion of patients receiving non-invasive ventilation support, rising from 12% to 18%. This surge in therapeutic interventions was notably mirrored by an increase in the percentage of patients treated at the ED’s own Intensive Care Unit (ICU), surging from 39% to 64%.

These observed shifts in treatment patterns likely represent a confluence of factors, including an influx of more severe cases, as indicated by the distinctly heightened one-year mortality rates—a trend aligning with findings in other studies [18,19]. Additionally, alterations in management strategies may be at play, potentially influenced by a heightened “fear of aerosols”. This fear could explain the exclusive increase in therapy involving non-invasive ventilation support, utilizing tight face masks, without a corresponding rise in HFNC. The concern for potential transmission between patients due to insufficient isolation capacity in the general ED area might have prompted physicians to opt for broader transfers to the ED’s ICU. The stability in the transfer rates to other ICUs after initial ED treatment could be considered supportive evidence for this hypothesis.

Healthcare provision during the pandemic undoubtedly bore the imprint of numerous influences. Medical considerations, such as the perceived severity of cases and evolving knowledge about COVID-19, likely played a pivotal role in guiding treatment decisions. Concurrently, non-medical factors, including the prevailing “fear of aerosols” and concerns about transmission risks within the healthcare setting, may have exerted a notable influence. These factors collectively highlight the complex decision-making processes that healthcare providers navigated, balancing the obligation to provide effective care with the necessity to mitigate potential transmission risks.

In summary, our study underscores the complex relationship of many medical and non-medical factors that have left a permanent mark on patient numbers and management practices during the pandemic. Figure 5 provides a visual representation, illustrating potential interactions between the various phenomena encountered in the complicated healthcare landscape during this unprecedented period.

### Limitations

This was a before–after study with all the limitations inherent to this design. In particular, no causal inference should be drawn from the findings. The nature of this study design restricts our interpretation to a description of observed changes. For instance, while our study sheds light on notable shifts in therapy choices, such as the potential influence of a “fear of aerosols”, we must underscore the inability to conclusively prove the reasons behind providers opting for one therapeutic approach over another.

Furthermore, the scope of our investigation was specifically tailored to patients treated for AECOPD within the emergency department (ED), thereby confining the generalizability of our findings to this specific clinical setting. The outcomes and dynamics observed in our study might not necessarily be extrapolated to other medical contexts or patient populations. Additionally, the study was conducted as a single-center study, although within the confines of a large tertiary care center. While this singular focus allows for a detailed exploration within a specific institutional framework, physicians have to be cautious when extending these findings to a broader healthcare landscape.

We nevertheless believe that our findings carry meaningful implications, particularly in shedding light on the challenges faced by an especially vulnerable group of patients—those undergoing treatment for AECOPD—during the unprecedented era of the COVID-19 pandemic. This population’s unique vulnerabilities and healthcare needs warrant focused attention, and our study may contribute valuable insights into the nuanced changes observed in their care within the ED setting. It is essential to contextualize our study within the broader context of existing research because the impact of the pandemic on the general ED population has been extensively studied. By focusing on this specific subset of patients, our study addresses a crucial gap in the literature, providing a better understanding of the challenges and adaptations encountered in the care of individuals grappling with AECOPD amidst the evolving landscape of the COVID-19 pandemic.

## 5. Conclusions

In the complex field of healthcare dynamics during the COVID-19 pandemic, a noticeable and statistically significant decline was observed in the overall number of patients seeking treatment for AECOPD. Our findings highlight some of the changes that occurred in patient flow into the ED. Within a diminished patient cohort, a compelling shift in clinical acuity emerged, with patients manifesting a more severe illness profile. This heightened severity manifested in the augmented utilization of NIV and a notably discernible increase in one-year mortality rates. These nuanced shifts underscore the multifaceted challenges faced by both patients and healthcare providers within the evolving landscape of the pandemic. It becomes evident that the healthcare providers’ decisions were likely shaped by a complex interplay of diverse medical and non-medical factors. The observed increase in the employment of NIV as a therapeutic modality during the pandemic, coupled with the elevated one-year mortality rates, suggests a distinctive pattern that necessitates a more comprehensive exploration. While the observed trends are suggestive, drawing conclusive links between specific factors and treatment choices requires a more detailed examination.

## Figures and Tables

**Figure 1 healthcare-12-00637-f001:**
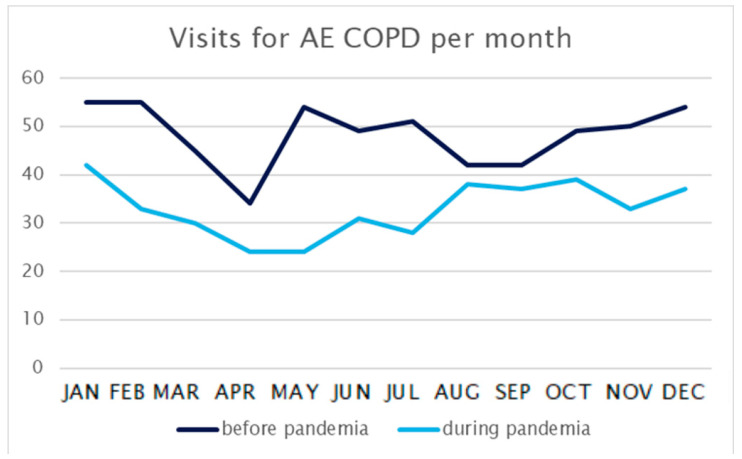
Visits to the emergency department for AECOPD per month.

**Figure 2 healthcare-12-00637-f002:**
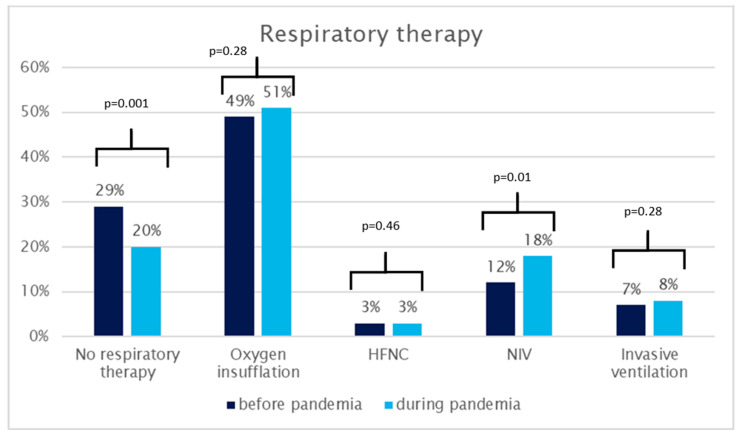
Respiratory therapy. HFNC: high-flow nasal cannula, NIV: non-invasive ventilation support.

**Figure 3 healthcare-12-00637-f003:**
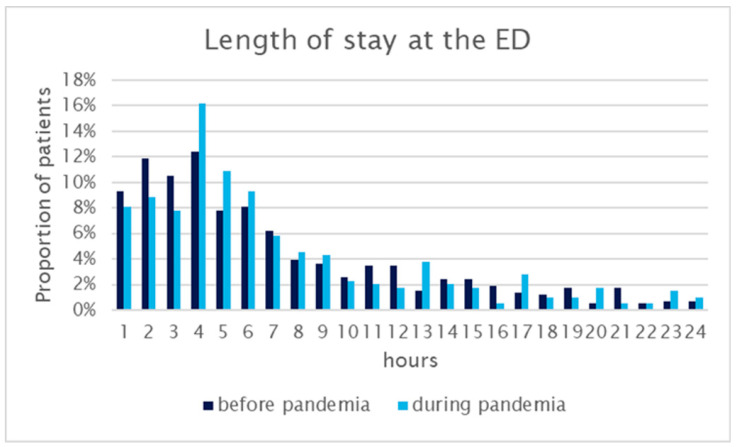
Length of stay (emergency department including emergency department-ICU).

**Figure 4 healthcare-12-00637-f004:**
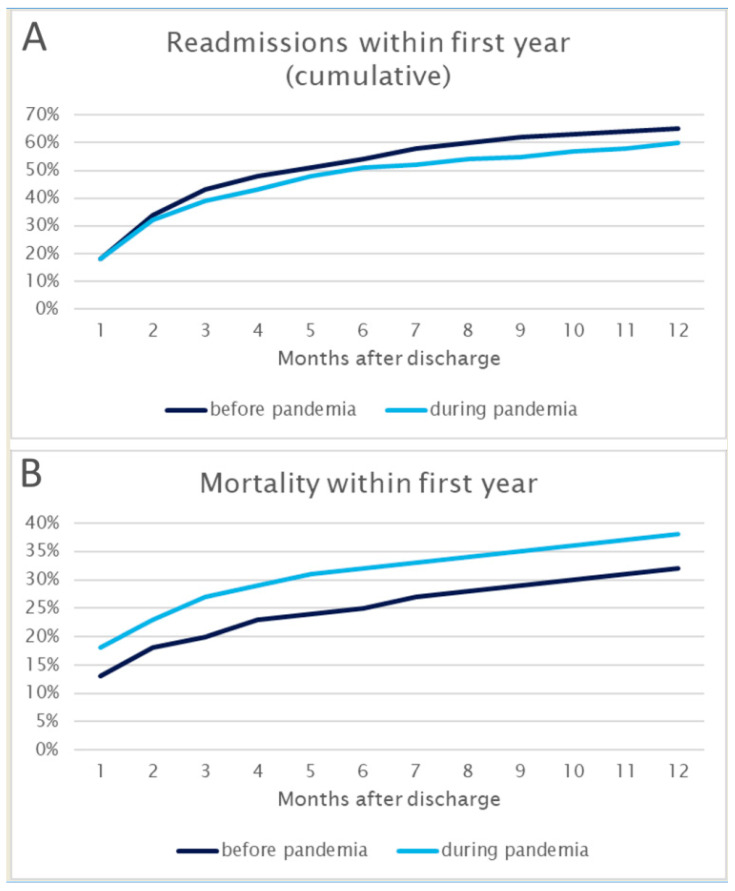
Re-admissions (Panel (**A**)) and overall mortality (Panel (**B**)) per month within the first year after presentation to the emergency department.

**Figure 5 healthcare-12-00637-f005:**
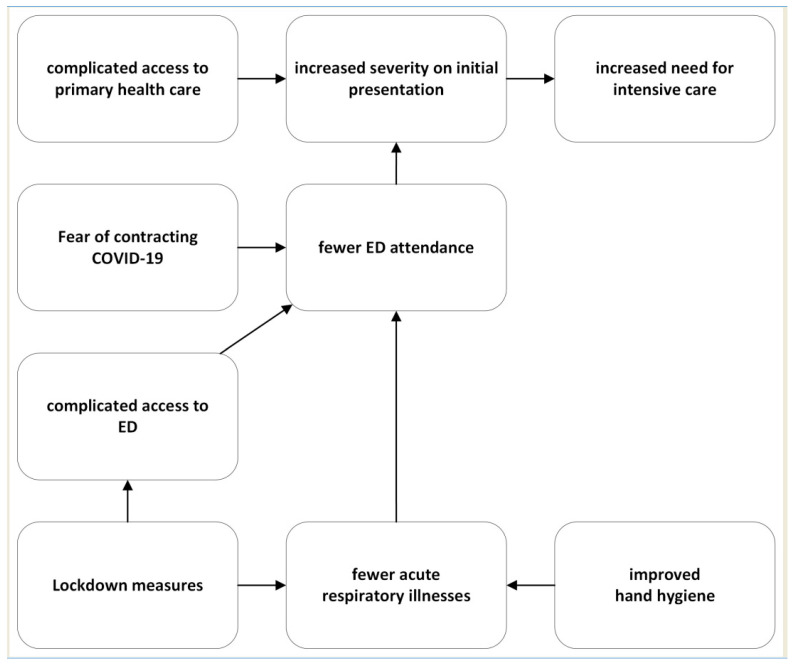
Visualized concept for potential interactions between the various phenomena influencing the number of patients at the emergency department associated with the COVID-19 pandemic.

**Table 1 healthcare-12-00637-t001:** Patient characteristics, vital signs, and laboratory values at first presentation. All values are means and standard deviations unless stated otherwise. CRP: c-reactive protein; IQR: inter-quartile range.

	Before Pandemic (n = 580)	During Pandemic (n = 396)	Effect Size Differences (95% Cis)
Age (years; median; IQR)	70 (63–77)	72 (65–79)	+1.5 (0.2–2.9)
Gender (male; n; %)	295 (51%)	218 (55%)	+4% (−2–11%)
Long-term oxygen therapy (n; %)	177 (30%)	132 (33%)	+3% (−3–9%)
Blood pressure systolic (mmHg)	137 (56)	140 (87)	+9 (−1–19)
Blood pressure diastolic (mmHg)	80 (18)	81 (18)	+1 (−1–4)
Heart rate (/min)	97 (24)	97 (25)	+/−0 (−3–4)
pCO_2_ (mmHg)	50 (23)	52 (23)	+4 (−0.5–8)
pH	7.4 (0.1)	7.3 (0.2)	−0.3 (−0.8–0.2)
Lactate (mmol/L)	1.9 (3.1)	2.4 (2.2)	−0.2 (−0.3–0.7)
Leucocytes (G/L)	11.1 (5.6)	11.9 (5.9)	−4 (−4–12)
CRP (mg/dL)	2.0 (7.5)	1.9 (6.5)	+0.4 (−0.3–1.1)
Serum sodium (mmol/L)	135 (11)	136 (8)	+1 (−2–3)
Serum potassium (mmol/L)	4.7 (0.6)	4.4 (0.8)	−0.3 (−1.6–0.9)
Creatinine (mg/dL)	1.5 (3.7)	1.4 (1.2)	−0.1 (−0.5–0.3)
Blood urea nitrogen (mg/dL)	25 (18)	28 (22)	+3 (0.1–5.3)
Antibiotics (n; %)	107 (18%)	91 (23%)	+5% (−1–10%)
Emphysema (n; %)	47 (8%)	49 (12%)	+4% (0–8%)
Pulmonary venous congestions (n; %)	82 (14%)	96 (24%)	+10% (5–15%)
Atrial fibrillation (n; %)	36 (6%)	54 (14%)	+7% (4–11%)

## Data Availability

Data are contained within the article.

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
