# Peer review of "Impact of the COVID-19 Pandemic on Therapy and Outcome of Acute Exacerbations of Chronic Obstructive Lung Disease at the Emergency Department"

_healthcare, 2024, doi:10.3390/healthcare12060637_

Round 1

Reviewer 1 Report

Comments and Suggestions for Authors

Dear Authors this study is well designed and interseting for making some conclusions for future decision making.

Although it is a retrospective chart review it gives strength evidence about reduction in ED visits during pandemic. Also, the increase in the percentage of patients treated at the ED's own Intensive Care Unit (ICU) was an important result of your study. I have only 2 comments.

1. Reading the title, I expected that you had searched also COVID-19 E.D. patients, but finally they were only ten (low, medium or large? give us the literature if exists). So, you didn't make any statistical tests comparing to COVID-19 positive Vs non-COVID-19 patients reagrding the whole management? How was the management in the pandemic year between these 2 groups?

2. Please clarify in the text, more precisely the Figure 5, explaining the effect of the arrows (up-down) as resulting to something?

Best Regards,

Reviewer 2 Report

Comments and Suggestions for Authors

In this comprehensive study comparing emergency department (ED) visits for Acute Exacerbation of Chronic Obstructive Pulmonary Disease (AECOPD)  before and during the initial wave of the COVID-19 pandemic Fuhrmann  et al report a pronounced reduction in ED visits and increased need for non-invasive MV in ED. Overall, the manuscript is well written.  Although reduction in AECOPD during COVID is well-known this study offers unique insight into flow of these patients in ED. This should be highlighted in the manuscript. Apart from that I only have following comments:

1. The authors should consider using ordinal  WHO scale for respiratory support as an outcome

2. Figure 5 is difficult to read. It should be redrawn to make it more clear.

Comments on the Quality of English Language

.

Reviewer 3 Report

Comments and Suggestions for Authors

I read with interest the paper titled "Impact of the COVID-19 pandemic on therapy and outcome of acute exacerbations of chronic obstructive lung disease at the emergency department"

Methods: How does the study ensure the accuracy and reliability of the data collected, especially considering the challenges posed by the COVID-19 pandemic on data gathering and healthcare services?

Please describe clearly what statistical tests were performed and where. The sentence used is very broad and general "We compared the “before” and “pandemic” groups using standard methods (chisquare test, t-test, Mann-Whitney-U-Test).". Where you performed chisquare? i didnt find any results. You present t-test and U-test (parametric and non-parametric) for means. Which tests did you use for the assumptions of normal data? (normality and equal variance??)

How was effect size differences calculated? Which type of effect size was measured? Please specify in the methods section. 

Present p-values on Table 1. 

Does the study's patient cohort accurately represent the larger population of patients with acute exacerbations of chronic obstructive pulmonary disease (AECOPD)? Please provide background on incidence per age groups and present your data as well divided by age groups. 

How does the study address potential biases in patient selection?

Minor comments:

Authors list: "Dominik ROTH and MD PhD" - please correct.

In table 1, you use zero one and two decimals. Pleas use the same amount of decimals in all units. 

Figure 2 - add p-values in each pair of bars. 

Acronyms are defined repeatedly during the paper. Please define once and then use the acronym during the text. eg: AECOPD is defined 4 times in the main text. 

Round 2

Reviewer 3 Report

Comments and Suggestions for Authors

The authors adressed most of the questions. Nothing to add. Accept in the current form.